# Global distribution and drivers of relative contributions among soil nitrogen sources to terrestrial plants

Chao-Chen Hu[1], Xue-Yan Liu [1] ✉, Avery W. Driscoll [2], Yuan-Wen Kuang[3], E. N. Jack Brookshire [4], Xiao-Tao Lü [5], Chong-Juan Chen[1], Wei Song[1], Rong Mao[6], Cong-Qiang Liu [1] & Benjamin Z. Houlton[7]

Soil extractable nitrate, ammonium, and organic nitrogen (N) are essential N sources supporting primary productivity and regulating species composition of terrestrial plants. However, it remains unclear how plants utilize these N sources and how surface-earth environments regulate plant N utilization. Here, we establish a framework to analyze observational data of natural N isotopes in plants and soils globally, we quantify fractional contributions of soil nitrate ($f_{NO3-}$), ammonium ($f_{NH4+}$), and organic N ($f_{EON}$) to plant-used N in soils. We find that mean annual temperature (MAT), not mean annual precipitation or atmospheric N deposition, regulates global variations of $f_{NO3-}$, $f_{NH4+}$, and $f_{EON}$. The $f_{NO3-}$ increases with MAT, reaching 46% at 28.5 °C. The $f_{NH4+}$ also increases with MAT, achieving a maximum of 46% at 14.4 °C, showing a decline as temperatures further increase. Meanwhile, the $f_{EON}$ gradually decreases with MAT, stabilizing at about 20% when the MAT exceeds 15 °C. These results clarify global plant N-use patterns and reveal temperature rather than human N loading as a key regulator, which should be considered in evaluating influences of global changes on terrestrial ecosystems.

Nitrogen is a vital nutrient element for life on Earth. Vascular plants dominate biomass and carbon (C) capture on land where N limitation is widespread[1]. Accordingly, a better understanding of plant N-use mechanisms is critical for assessing and predicting primary productivity of terrestrial ecosystems[2,3]. Global changes such as climate warming and increasing atmospheric N deposition have significantly impacted the soil N cycle and plant N utilization and consequently terrestrial primary productivity[4,5]. Nevertheless, the exact contributions of soil N sources to terrestrial plants (i.e., how plants utilize the available soil N sources) remain unquantified and their variations

among global terrestrial environments remain unclear[6]. This knowledge gap is preventing an accurate evaluation of N-cycle effects on biodiversity and the C cycle, as well as their responses to projected environmental changes[3,7].

Non-$N_2$-fixing plants are assumed to primarily acquire bioavailable N from soils via roots[1]. The total extractable N (TEN) pool accessible to microbes and plants includes nitrate ($NO_3^-$), ammonium ($NH_4^+$), and organic N (EON) (Fig. 1)[3,6,8]. Plant roots acquire soil extractable N directly or via mycorrhizal associaton[9]. Acquired N is allocated and assimilated among leaves, stems, and roots, which combined

[1]School of Earth System Science, Tianjin University, Tianjin, China. [2]Department of Soil and Crop Sciences, Colorado State University, Fort Collins, CO, USA. [3]Guangdong Provincial Key Laboratory of Applied Botany and Key Laboratory of Vegetation Restoration and Management of Degraded Ecosystems, South China Botanical Garden, Chinese Academy of Sciences, Guangzhou, China. [4]Department of Land Resources and Environmental Sciences, Montana State University, Bozeman, MT, USA. [5]Erguna Forest-Steppe Ecotone Research Station, Institute of Applied Ecology, Chinese Academy of Sciences, Shenyang, China. [6]Key Laboratory of National Forestry and Grassland Administration On Forest Ecosystem Protection and Restoration of Poyang Lake Watershed, College of Forestry, Jiangxi Agricultural University, Nanchang, China. [7]Department of Global Development and Department of Ecology and Evolutionary Biology, Cornell University, Ithaca, NY, USA. ✉e-mail: liuxueyan@tju.edu.cn

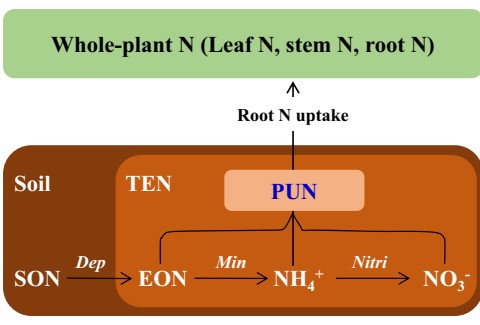

**Fig. 1 | Sources and processes of PUN in terrestrial ecosystems.** PUN plant-used N, SON soil organic N, TEN total extractable N, EON extractable organic N, $NH_4^+$ ammonium, $NO_3^-$ nitrate, Dep depolymerization, Min mineralization, Nitri nitrification.

constitutes the whole-plant N pool (Fig. 1)[10,11]. Before the 1980s, soil inorganic N was recognized as the exclusive plant N source, and thus soil N mineralization has long been used to evaluate the plant-used N in soils (PUN; Fig. 1)[12]. Nevertheless, subsequent findings of root absorption of EON molecules (such as amino acids, peptides, proteins, and even microbes) in ecosystems under different climate contexts indicate that soil EON is a non-negligible contributor of PUN and soil N mineralization could not fully account for PUN[13-15]. Since then, the concentrations and pool sizes of $NO_3^-$, $NH_4^+$, and EON have been measured and used in combination with plant C/N ratios to model the plant-soil N and C cycle[6,7,16]. However, parallel evidence on the complex N competition between plants and microbes and intrinsic N preference of plants among $NO_3^-$, $NH_4^+$, and EON indicates that TEN cannot be simply taken as PUN[8,12]. For example, specific chemical fractions of the TEN pool that are immobilized and transformed by soil microbes can differ substantially from those observed in soil TEN[17]. Also, plant acquisition of soil $NO_3^-$, $NH_4^+$, and EON do not follow their proportions in soil TEN because of the verified plant N preference[18-20]. These two factors result in different estimates of pool sizes and chemical proportions between TEN and PUN in soils. Among the existing studies, the $^{15}$N isotope dilution method is effective at partitioning contributions of $NO_3^-$, $NH_4^+$, and EON to PUN, respectively[8]. However, the high cost of $^{15}$N tracers and artificial injection makes it most applicable for incubating plants and small-scale experiments. Further, asynchrony in N species, time, and space among short-term $^{15}$N additions, microbial turnover, and root absorption caused substantial uncertainties in evaluating the integrative long-term mechanisms of plant N utilization (Table s1)[8,12]. For natural ecosystems, the $^{15}$N-tracer application with water can strongly alter soil chemistry and disturb microbe-soil-plant N relationships[13,14]. For EON, the $^{15}$N tracer is restricted to few molecules (e.g., free amino acids, which only account for <5% of EON in soil[21]) and thus cannot accurately elucidate plant EON utilization. Therefore, an integrative and non-invasive approach is strongly needed for deciphering 'in-situ' N-use mechanisms of terrestrial plants.

The ratio of natural N isotopes (i.e., $^{15}$N/$^{14}$N, denoted as $\delta^{15}$N values and expressed in per mille units; $\delta^{15}$N = [($^{15}$N/$^{14}$N)$_{sample}$ / ($^{15}$N/$^{14}$N)$_{standard}$] − 1] × 1000, (where the standard is atmospheric $N_2$) has been known as a non-invasive measure to decipher N-cycle processes[22-27]. Hitherto, the combination of leaf $\delta^{15}$N (denoted as $\delta^{15}$N$_{leaf}$) with the $\delta^{15}$N of soil $NO_3^-$, $NH_4^+$, and EON (denoted as $\delta^{15}$N$_{soil-NO3}$, $\delta^{15}$N$_{soil-NH4+}$, and $\delta^{15}$N$_{soil-EON}$, respectively) and Bayesian isotope mixing models has been recognized as a feasible method to quantify respective contributions of soil $NO_3^-$, $NH_4^+$, and EON to PUN[18-20]. However, two fundamental questions concerning the quantification of soil N sources to PUN remain unresolved at the global scale. First, the intra-plant $^{15}$N heterogeneity causes differences between the whole-plant $\delta^{15}$N (denoted as $\delta^{15}$N$_{plant}$) and $\delta^{15}$N$_{leaf}$[28]. This difference caused errors in interpreting plant N sources by using $\delta^{15}$N$_{leaf}$ but has

not been constrained to characterize $\delta^{15}$N$_{plant}$ signatures[18]. Additionally, the preferential uptake and transformation of $^{14}$N from mycorrhiza to plants cause lower $\delta^{15}$N$_{plant}$ than the corresponding $\delta^{15}$N of PUN (denoted as $\delta^{15}$N$_{PUN}$)[24,26]. The discrepancies between $\delta^{15}$N$_{plant}$ and $\delta^{15}$N$_{PUN}$ ($\Delta_m$) not only differ among mycorrhiza types, but also vary with environmental conditions influencing the dependence of plant N uptake on mycorrhiza[9]. However, no study has constrained $\Delta_m$ values for specific mycorrhizal symbioses across contrasting environments, thus hampering understanding of variation in $\delta^{15}$N$_{PUN}$ of global terrestrial plants.

Second, it is a long-standing question what proportions of soil N sources contribute to PUN under different environmental backgrounds, so that the environmental control mechanisms of the fractional variations remain unclear. Regarding the contributing proportions, some studies showed that plants under lower MAT (<5 °C) or at higher latitudes (> 63 °N) mainly utilized EON (43–66% (c.a. >59%) for tundra plants) (data compiled in Table s1). In contrast, the other studies in these regions estimated much lower contributions of organic N (c.a. <22%), with c.a. 14–61% and 24–63% from $NH_4^+$ and $NO_3^-$, respectively (Table s1). Similarly, some studies showed that plants under higher MAT (>12 °C) or at lower latitudes (<38 °N) mainly used inorganic N (c.a. 86–95%), but other studies in these regions argued that the contribution of organic N to PUN reached 20-39% (Table s1). These contrasting findings demonstrate that the proportional contributions of soil N sources to PUN remain an open question. Regarding the environmental drivers, global change studies have confirmed that increasing atmospheric N deposition, temperature, and precipitation are three main factors affecting ecosystem N cycling[6]. However, existing studies have been mostly based on simulations of single or two factors and mainly concerned plant inorganic N utilization[29]. In the 'real' world of global terrestrial ecosystems, it remains uncertain whether and how these factors influence the geographic distribution of the relative contributions of soil $NH_4^+$, $NO_3^-$, EON to PUN. Based on higher soil inorganic N concentrations, mineralization and nitrification rates under simulated warming and N additions[17,30], plant inorganic N and $NO_3^-$ utilization were assumed to increase with increasing temperature and N deposition[31]. However, based on a data compilation of sparse observations (data compiled in Table s2), only a temperature effect was observed on plant inorganic N uptake. For precipitation, an observed phenomenon is that the contribution of $NH_4^+$ relative to $NO_3^-$ to PUN increased with mean annual precipitation (MAP) because of inhibition of nitrification and enhanced denitrification[18,29]. However, due to variation in experimental conditions it remains uncertain which environmental factors are regulating soil N source contributions to PUN and how these contributions change across terrestrial biomes.

Here, this study resolves the long-standing question of natural N abundance isotope methods for constraining $\delta^{15}$N$_{PUN}$ signatures and accomplishes the quantification of soil $NO_3^-$, $NH_4^+$, and EON contributions to PUN and their global patterns, respectively. First, we update the global $\delta^{15}$N$_{leaf}$ dataset based on that in Craine et al.[26] and the literature published since January 10$^{th}$, 2018 (Figs. s1a, s2a; Supplementary Text 1). By compiling the global data of $\delta^{15}$N$_{leaf}$, stem $\delta^{15}$N ($\delta^{15}$N$_{stem}$), and root $\delta^{15}$N ($\delta^{15}$N$_{root}$) measured for the same plant individuals (Figs. s1b, s2; Supplementary Text 2), we establish the relationship between $\delta^{15}$N$_{leaf}$ and $\delta^{15}$N$_{plant}$ to constrain the corresponding $\delta^{15}$N$_{plant}$ values of global $\delta^{15}$N$_{leaf}$ observations (Fig. s3). Then, we analyze the effects of MAT, MAP, and plant life form on $\delta^{15}$N$_{plant}$ of different mycorrhizal plants (Table s3, Fig. s4) and establish the relationships of $\delta^{15}$N$_{plant}$ with MAT for plants with the same mycorrhizal type and life form (Fig. s5). Using these relationships, we constrain $\Delta_m$ values for global $\delta^{15}$N$_{plant}$ values and finally obtain the corresponding $\delta^{15}$N$_{PUN}$ signatures (Table s4). Further, we compile a global dataset of $\delta^{15}$N$_{soil-NO3}$, $\delta^{15}$N$_{soil-NH4+}$, and $\delta^{15}$N$_{soil-EON}$ (Figs. s1c, s6; Supplementary Text 3), analyze the effects of climate and soil parameters on these source $\delta^{15}$N values of PUN (Table s5), establish their

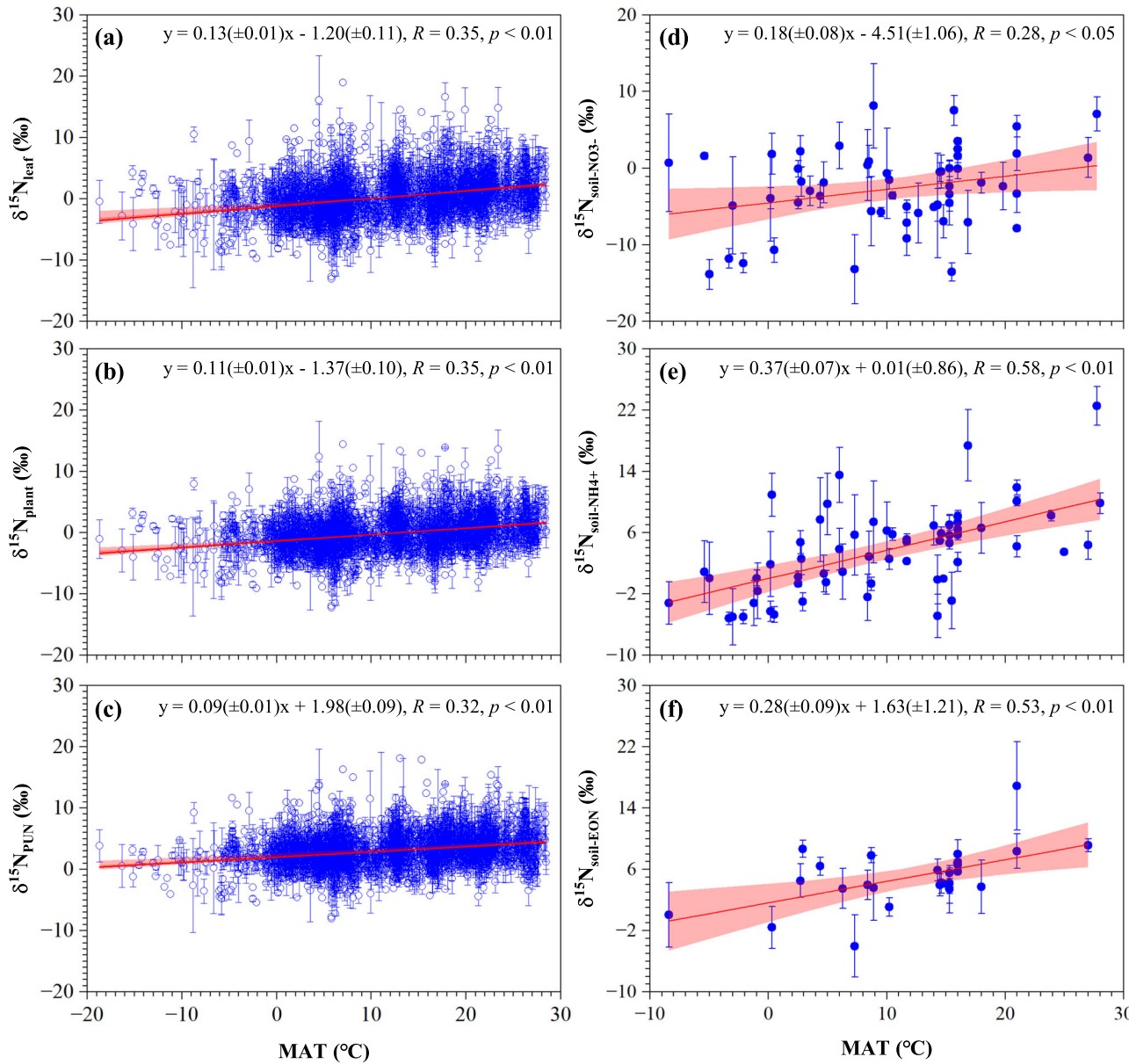

**Fig. 2 | Nitrogen isotope ratios of plants and soils.** Variations of site-based mean $\delta^{15}N$ of leaf N ($\delta^{15}N_{leaf}$), whole-plant N ($\delta^{15}N_{plant}$), PUN ($\delta^{15}N_{PUN}$), soil $NO_3^-$ ($\delta^{15}N_{soil-NO3-}$), soil $NH_4^+$ ($\delta^{15}N_{soil-NH4+}$), and soil EON ($\delta^{15}N_{soil-EON}$) with MAT are shown in panels (**a**)–(**f**), respectively. Mean and SD values were based on sample replicates at each site ($n = 1$–2118 for plants, and $n = 2$–21 for soils). Site distribution is shown in Fig. s1. The regression was analyzed by fitting effects with 95% confidence intervals.

relationships with MAT (Fig. 2) to constrain site-based mean $\delta^{15}N_{soil-NO3-}$, $\delta^{15}N_{soil-NH4+}$, and $\delta^{15}N_{soil-EON}$ values for the corresponding site-based $\delta^{15}N_{PUN}$ values (Fig. 2). Based on site-based $\delta^{15}N_{PUN}$ and source $\delta^{15}N$ values in each grid cell (0.1° (latitude) × 0.1° (longitude)) and using the Stable Isotope Analysis in R (i.e., the SIAR model) (detailed in Methods), we accomplish the calculations of fractional contributions of soil $NO_3^-$, $NH_4^+$, and EON to PUN for global 1610 grid cells (Fig. s7). Finally, we examine relationships between the PUN source contributions and major environmental factors to evaluate the environmental controls of plant N utilization across terrestrial ecosystems.

## Results

### $\delta^{15}N$ values of leaf, stem, root, and the whole plant

The $\delta^{15}N_{leaf}$, $\delta^{15}N_{stem}$, and $\delta^{15}N_{root}$ of terrestrial plants showed substantial differences from each other ($p < 0.05$, Fig. s2a–c). Based on the same plant individuals, $\delta^{15}N_{leaf}$ was higher by $0.5 \pm 2.5$‰ than $\delta^{15}N_{root}$ and by $0.4 \pm 1.2$‰ than $\delta^{15}N_{stem}$ (Fig. s2d, e), positive correlations

between $\delta^{15}N_{leaf}$ and $\delta^{15}N_{root}$ or $\delta^{15}N_{stem}$ showed slope values of 0.57 and 0.90, respectively (Fig. s3). There was a linear correlation between $\delta^{15}N_{leaf}$ and the corresponding $\delta^{15}N_{plant}$ for herbs, shrubs, and trees (Fig. s3c), which were used to calibrate global $\delta^{15}N_{leaf}$ observations to the corresponding $\delta^{15}N_{plant}$ values. Globally, $\delta^{15}N_{plant}$ values differed distinctly among herb, shrub, and tree (Table s3). Moreover, $\delta^{15}N_{plant}$ values of the same life form differed among mycorrhizal types, showing a general order of nonmycorrhizal (NM) > arbuscular mycorrhizal (AM) > ectomycorrhizal (ECM) > ericoid mycorrhizal (ERM) (Fig. s4). For the same life form and mycorrhizal type, $\delta^{15}N_{plant}$ values varied linearly with the MAT (Fig. s5), which was used to calculate the corresponding $\delta^{15}N_{PUN}$ values (Table s4).

### $\delta^{15}N$ values of PUN and soil N sources

The $\delta^{15}N_{PUN}$ values averaged $3.4 \pm 3.3$‰ ($-15.0$‰–21.7‰) (Fig. s6a) and increased linearly with MAT ($\delta^{15}N_{PUN} = 0.09 \times MAT + 1.98$; Fig. 2). Similarly, soil $\delta^{15}N_{soil-NO3-}$, $\delta^{15}N_{soil-NH4+}$, and $\delta^{15}N_{EON}$ values were also

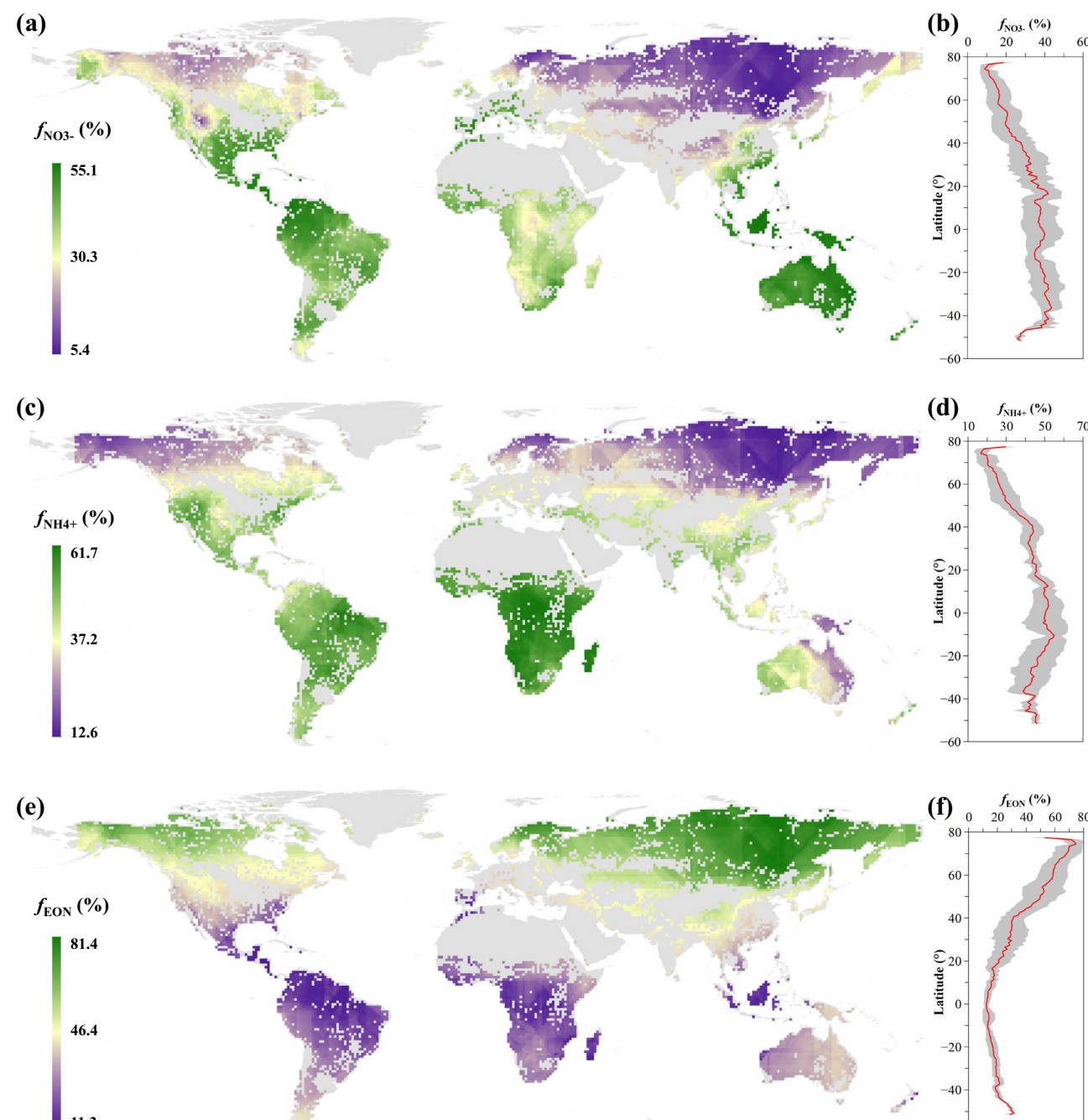

**Fig. 3 | Geographical distributions and latitudinal variations of soil N-source contributions to terrestrial plants.** $f_{NO3-}$ (**a**, **b**), $f_{NH4+}$ (**c**, **d**), and $f_{EON}$ (**e**, **f**) are fractional contributions of soil $NO_3^-$, $NH_4^+$, and EON to PUN, respectively. In panels (**a**), (**c**), and (**e**), gray areas are deserts, glaciers, perennial snow-covered areas, bare ground, and agricultural land etc. The mapping was conducted by ArcGIS version 10.8 (Esri Inc., USA) using the Kriging spatial interpolations based on the 0.1° (latitude) × 0.1° (longitude) grid-based data. The base map was downloaded from https://hub.arcgis.com/datasets/esri::world-countries-generalized. In panels (**b**), (**d**), and (**f**), red lines and gray areas show mean and SD values.

---

mainly influenced by MAT (Table s5) and increased linearly with MAT (Fig. 2), showing different slope and intercept values ($\delta^{15}N_{soil-NO3-} = 0.18 \times MAT - 4.51$, $\delta^{15}N_{soil-NH4+} = 0.37 \times MAT + 0.01$, and $\delta^{15}N_{EON} = 0.28 \times MAT + 1.63$; Fig. 2) from that of $\delta^{15}N_{PUN}$. For sites with simultaneous N concentration and isotope observations in both plant and soil, the site-based mean $\delta^{15}N$ values of TEN ($\delta^{15}N_{TEN}$) ($5.4 \pm 2.7‰$) were higher by $2.3 \pm 2.7‰$ on average than the corresponding $\delta^{15}N_{PUN}$ ($3.1 \pm 2.6‰$) and the differences generally increased with MAT (Fig. s8).

**Fractional contributions of soil $NO_3^-$, $NH_4^+$, and EON to PUN**

The plant $f_{NO3-}$, $f_{NH4+}$, and $f_{EON}$ values averaged $29 \pm 19\%$, $42 \pm 18\%$, and $29 \pm 19\%$, respectively (Fig. s7). Generally, the $f_{NO3-}$ and $f_{NH4+}$ increased while the $f_{EON}$ decreased with the latitude (Figs. 3, s9). Neither plant $f_{NO3-}$, $f_{NH4+}$, nor $f_{EON}$ values showed a clear relationship with MAP and the flux of atmospheric N deposition, instead they varied with the MAT nonlinearly (Figs. 4, s10). Across the observed temperature spectrum, the $f_{NO3-}$ increases with MAT and reaches a peak of 46% at 28.5 °C ($f_{NO3-} = 0.01 \times MAT^2 + 0.65 \times MAT + 20.8$; Fig. 4a). The $f_{NH4+}$ also

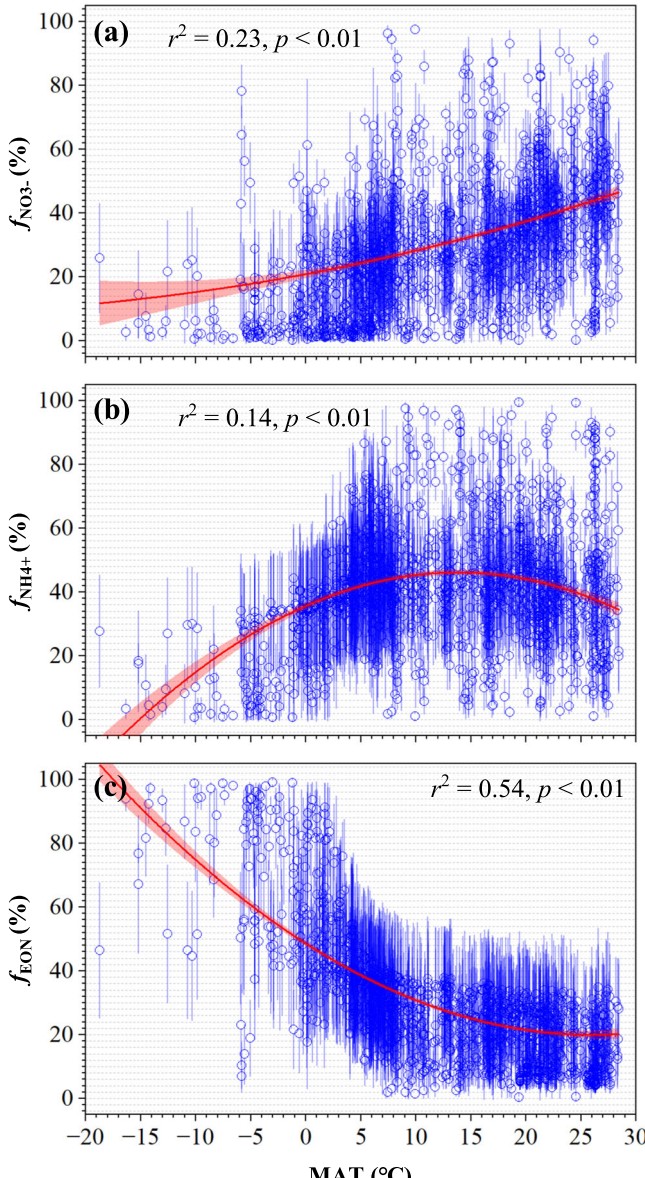

**Fig. 4 | Variations of soil N-source contributions to PUN with MAT.** $f_{NO3-}$ (**a**), $f_{NH4+}$ (**b**), and $f_{EON}$ (**c**) are fractional contributions of soil $NO_3^-$, $NH_4^+$, and EON to PUN, respectively. The 0.1° (latitude) × 0.1° (longitude) grid-based mean ± SD values are shown. The regression was analyzed by fitting effects with 95% confidence intervals.

increased with MAT, achieving a maximum of 46% at 14.4 °C, followed by a decline as temperatures further increased ($f_{NH4+} = -0.05 \times MAT^2 + 1.52 \times MAT + 35.6$; Fig. 4b). Meanwhile, the $f_{EON}$ gradually decreased with MAT, stabilizing at about 20% when the MAT exceeded 15 °C ($f_{EON} = 0.04 \times MAT^2 - 2.20 \times MAT + 48.6$; Fig. 4c).

## Discussion

Differing $\delta^{15}N_{leaf}$, $\delta^{15}N_{stem}$, and $\delta^{15}N_{root}$ of terrestrial plants at both site (Fig. s2a–c) and individual levels (Fig. s2d–f) demonstrated substantial N isotope effects of intra-plant N assimilation/allocation[28,32,33] and none of them can exactly represent the corresponding $\delta^{15}N_{plant}$ signatures[28,34]. The generally higher $\delta^{15}N_{leaf}$ than the corresponding $\delta^{15}N_{plant}$ (Figs. 2a, b, s3c) points to a potential risk of overestimating the contribution of $^{15}N$-enriched soil N sources to terrestrial plants when neglecting the intra-plant N isotope effects on $\delta^{15}N_{leaf}$ values.

Accordingly, the $\delta^{15}N_{leaf}$ vs $\delta^{15}N_{plant}$ correlation based on the same plant individuals (Fig. s3c) provides a transformation of global $\delta^{15}N_{leaf}$ measurements (Fig. 2a) to the corresponding $\delta^{15}N_{plant}$ values (Fig. 2b).

The generally increasing $\delta^{15}N_{plant}$ of the same life form and mycorrhizal type with MAT (Fig. s5) revealed temperature as an effective predictor of mycorrhizal N isotope effects on $\delta^{15}N_{plant}$ on the global scale (Table s4). The decreasing mycorrhizal N isotope effects with increasing MAT for most plants (Table s4) confirmed a weaker mycorrhizal mediation of the whole-plant N acquisition under warmer climate conditions[9,24]. This tendency might be related to the generally increasing soil bioavailable N (particularly inorganic N sources) with MAT for the direct N absorption via plant roots[9,24]. The correlation between mycorrhizal N isotope effects and MAT for the same life form and mycorrhizal type identified in our study (Table s4) allowed a transformation of global $\delta^{15}N_{plant}$ measurements (Fig. 2b) to the corresponding $\delta^{15}N_{PUN}$ values in soils (Fig. 2c).

Globally, $\delta^{15}N_{soil-TEN}$ (Fig. s8b), $\delta^{15}N_{soil-NO3-}$, $\delta^{15}N_{soil-NH4+}$, and $\delta^{15}N_{soil-EON}$ all increased linearly with MAT (Fig. 2d–f). Increasing temperature generally enhances rates of microbial N mineralization, nitrification, and denitrification in soils[30]. The openness of the soil N cycle would further increase the production and losses of $^{15}N$-depleted N species, such as the N-containing gases (e.g., NO, $N_2O$, $N_2$, $NH_3$)[22,25,35]. This mechanism was well supported by increasing $^{15}N$ abundances of soil bulk N pools with MAT[25,36]. Because the N nutrition of terrestrial plants depends on acquiring $NO_3^-$, $NH_4^+$, and EON from soils, the $\delta^{15}N_{PUN}$ naturally followed soil N sources to increase with MAT (Fig. 2c). These results demonstrate a temperature-induced openness of the soil N cycle[25] and offer $\delta^{15}N_{PUN}$ as an indicator of the plant-soil N cycle.

Based on simultaneous N concentration and isotope observations in both plants and soils (Figs. s8, s11–12), we confirmed distinctly lower $\delta^{15}N$ values in PUN than in TEN (Fig. s8). Clearly, neither $\delta^{15}N_{plant}$ nor $\delta^{15}N_{PUN}$ can exactly represent soil $\delta^{15}N_{TEN}$ to elucidate the soil or ecosystem bioavailable N, also soil $\delta^{15}N_{TEN}$ cannot be simply taken as $\delta^{15}N_{PUN}$. These results reflected differing sizes and compositions between the total bioavailable N and PUN pools in soils (Fig. 1). The generally lower $\delta^{15}N_{PUN}$ than $\delta^{15}N_{TEN}$ in soils found in our study (Fig. s8c) demonstrated that plants utilized relatively more $^{15}N$-depleted $NO_3^-$ than $^{15}N$-enriched $NH_4^+$ and EON in soils (Fig. s12). Interestingly, we found that $\delta^{15}N_{PUN}$ increased by a lower magnitude than the $\delta^{15}N_{soil-TEN}$ with increase MAT (Fig. s8a, b), which caused an increasing $\delta^{15}N_{TEN-PUN}$ value with MAT (Fig. s8c). In principle, $\delta^{15}N_{PUN}$ values were controlled by both $\delta^{15}N$ values and fractional contributions of the corresponding soil $NO_3^-$, $NH_4^+$, and EON[18,20,23]. Because $\delta^{15}N$ values of soil N sources are mainly regulated by microbial N cycles[17,37] and they increase with MAT (Fig. 2d–f), the lower sensitivity of $\delta^{15}N_{PUN}$ than $\delta^{15}N_{TEN}$ to MAT (Fig. s8) indicates changing fractional contributions of soil $NO_3^-$, $NH_4^+$, and EON to PUN, i.e., changing plant N-use strategies with MAT.

Globally, plant $f_{NO3-}$, $f_{NH4+}$, and $f_{EON}$ values averaged 29%, 42%, and 29%, respectively (Fig. s7). Clearly, plants mainly utilized inorganic N, though organic N is an important N source for plants[6,13,14]. Moreover, significant changes in $f_{NO3-}$, $f_{NH4+}$, and $f_{EON}$ with MAT, rather than with MAP or atmospheric N deposition flux (Figs. 4, s10), indicate a global temperature-dependent plant N-use pattern. This suggests that N deposition and MAP do not significantly affect plant N utilization, whereas MAT plays a critical role in driving the geographical distribution of plant N-use patterns at the global scale. From polar to tropical regions, both $f_{NO3-}$ and $f_{NH4+}$ increased, and $f_{EON}$ decreased, with decreasing latitude (Figs. 3, s9) and the increase in MAT (Fig. 4). These results provide quantitative evidence confirming that plants in relatively colder climate conditions (<0 °C) utilized soil EON in higher proportions (>50%) than plants in relatively warmer climate (Fig. 4c). Fractional contributions of three N forms to PUN changed with MAT nonlinearly (Fig. 4), which revealed that plant utilization of soil $NO_3^-$,

 

$NH_4^+$, and EON differed in responsiveness and sensitivity between colder and warmer climates. In colder climates (MAT of −18.7 to −2.0 °C), plant utilization of soil $NO_3^-$ was proportionally low (below 20%), attributed to significantly reduced nitrification rates[6,19]. In contrast, in warmer climates at sites with MAT ranging from 11.4 to 28.5 °C, soil $NO_3^-$ contribution to PUN exceeded the global average (29%), reaching up to 46% (Fig. 4a). Additionally, the contribution of soil $NH_4^+$ to PUN increased with rising MAT, peaking at 46% at 14.4 °C (Fig. 4b) due to increased N mineralization rates[12,30]. However, $f_{NH4+}$ declined with MAT when exceeding 14.4 °C (Fig. 4b), likely due to enhanced nitrification rates[38] and the toxic effects of excessive $NH_4^+$, causing lower intracellular pH and ionic imbalance in plants[3].

Based on simultaneous N concentration and isotope observations in both plant and soil (Figs. s8, s11-12), we found that $f_{NO3-}$, $f_{NH4+}$, and $f_{EON}$ did not differ substantially from those based on non-synchronous observations, showing consistently low $\Delta f_{NO3-}$, $\Delta f_{NH4+}$, and $\Delta f_{EON}$ values of −1.1 ± 11.1%, 2.0 ± 9.3%, and −0.9 ± 9.1%, respectively (Fig. s11). Also, calculating results based on simultaneous observations confirmed a globally temperature-dependent plant N-use pattern, i.e., both $f_{NO3-}$ and $f_{NH4+}$ increased and $f_{EON}$ decreased with MAT (although not significantly, Fig. s11). Moreover, we found that both $\beta_{NO3-}$ and $\beta_{NH4+}$ increased while $\beta_{EON}$ decreased with the increasing MAT ($\beta$ represents the preference degree of plant $NO_3^-$, $NH_4^+$, or EON utilization, detailed in Methods, Fig. s12). This pointed to an important plant N-use strategy that plants in relatively colder climate conditions preferred soil organic N sources over inorganic N sources[13,15]. Conversely, plants in relatively warmer climates preferred soil inorganic N sources over organic N sources[18,20]. In general, plants display a clear plasticity of relative N preference in response to increased MAT.

In summary, this study presents a quantitative analysis on levels and spatial variation of soil N source contributions to global terrestrial plants, providing methods and evidence for evaluating plant N-use mechanisms and N cycling of terrestrial ecosystems. By explicitly constraining isotope effects of intra-plant N assimilation/allocation and mycorrhizal N acquisition of different plant life forms under different climates, we transferred the $\delta^{15}N_{leaf}$ to a parameter of $\delta^{15}N_{PUN}$ for elucidating plant N utilization. Substantially differing signatures between $\delta^{15}N_{PUN}$ and $\delta^{15}N_{TEN}$ informed that neither $\delta^{15}N_{leaf}$ nor $\delta^{15}N_{TEN}$ can be directly taken as an indicator of soil N availability or the relative availability of PUN to soil N supply. Globally, we found that variations of $\delta^{15}N_{PUN}$ and therefore soil N source contributions to terrestrial plants were temperature-dependent and nonlinear, showing increasing plant inorganic N utilization and relative preference while decreasing organic N utilization and preference with increasing MAT. Our finding revealed the important role of plant N-use strategies in regulating plant $\delta^{15}N$ records and their responses to warming climate. Besides, due to differing C costs among plant $NH_4^+$, $NO_3^-$, and EON assimilation, our finding aids further evaluation of the effects of plant N utilization on C cycles.

However, extant $\delta^{15}N$ observations on roots and stems were still sparse among terrestrial plants, and simultaneous observations on $\delta^{15}N$, N concentrations, and biomass for leaves, stems, and roots were even less. Particularly, $\delta^{15}N$ values of soil extractable N sources have seldom been observed together with those of plants, particularly for sites and ecosystems with relatively lower MAT. Further, $\delta^{15}N$ values of soil EON sources that were actually used by plants remain difficult to determine. These are substantial uncertainties in the established relationships and calculated results of this study. Although it might not influence the general patterns of soil N source contributions to global plants found in the present study, to widely conduct simultaneous and point-to-point observations on plant-soil $\delta^{15}N$ parameters would provide more precise estimates on plant N sources and availability for biogeochemical and earth system modeling efforts.

## Methods

### Data compilation of $\delta^{15}N_{leaf}$, $N_{leaf}$, and N deposition

We collected the $\delta^{15}N_{leaf}$ data published after January 10th, 2018, and combined them with the existing $\delta^{15}N_{leaf}$ dataset collected by Craine et al.[26]. Briefly, we searched literature published since 2018 on the Web of Science and Google Scholar with the terms "nitrogen isotope or $^{15}N$" and "leaf or foliar". The $\delta^{15}N_{leaf}$ data of (1) urban areas, (2) agricultural ecosystems, (3) non-control samples of manipulative experiments, (4) non-vascular plants, (5) fertilized plants, (6) semi-aquatic or aquatic plants, and (7) $N_2$-fixing plants were excluded. Data in figures of publications were extracted using the software of Web Plot Digitizer (Version 4.2, San Francisco, California, USA).

By August 1st of 2022, 37 publications (listed in Supplementary Text 1) with the required $\delta^{15}N_{leaf}$ data were available, and a total of 16494 observations at 1218 sites were newly added to the dataset collected by Craine et al.[26] (41669 observations at 5296 sites before January 10th of 2018). Sites of all $\delta^{15}N_{leaf}$ observations (58163 observations at 6514 sites) distribute between 54.5°S and 71.1°N (Figs. s1a, s2a), with the MAT spanning from −18.7 °C to 28.6 °C and MAP ranging from 50.1 mm to 6576.0 mm. The data of MAT and MAP were collected either from the original literature or cited from the climatic database at http://www.worldclim.org using the coordinate information. All $\delta^{15}N_{leaf}$ observations were conducted for 5632 plant species of 2172 herbs, 1156 shrubs, and 2304 trees. The sampling years of $\delta^{15}N_{leaf}$ observations range from 1876 to 2022, with 90% sampled after 1995.

The $N_{leaf}$ data were also collected from the corresponding literature if available. Besides, to examine atmospheric N deposition effects on plant and soil N variables, we collected the data of deposition fluxes of inorganic and organic N in wet and dry deposition based on the observations in 12 individual years during 1984 − 2016[39]. We used the coordinate information of plant or soil $\delta^{15}N$ observation sites in our study to match and extract the corresponding data and then calculated mean annual fluxes of total N deposition.

### Constraining terrestrial $\delta^{15}N_{plant}$ signatures

The whole-plant N level ($N_{plant}$) is mainly determined by leaf, stem, and root N levels, so we have the following mass-balance Eq. (1).

$$N_{plant} = N_{leaf} \times F_{leaf} + N_{stem} \times F_{stem} + N_{root} \times F_{root} \tag{1}$$

where $N_{leaf}$, $N_{stem}$, and $N_{root}$ are the N concentrations in the leaf, stem, and root, respectively; $F_{leaf}$, $F_{stem}$, and $F_{root}$ are their respective biomass percentages in the whole plant. Then, the $\delta^{15}N_{plant}$ can be expressed by the following $\delta^{15}N$ mass-balance Eq. (2).

$$\delta^{15}N_{plant} = \delta^{15}N_{leaf} \times (N_{leaf} \times F_{leaf} / N_{plant}) + \delta^{15}N_{stem} \\ \times (N_{stem} \times F_{stem} / N_{plant}) + \delta^{15}N_{root} \times (N_{root} \times F_{root} / N_{plant}) \tag{2}$$

where we assume that $F_{leaf} + F_{stem} + F_{root} = 1$.

The N concentration and $\delta^{15}N$ data of stems and roots were collected from the publications of $\delta^{15}N_{leaf}$ observations. In our calculations (Eqs. (1) and (2)), we only used the N concentration and $\delta^{15}N$ data simultaneously measured for the same plant individuals (Fig. s3). The biomass data were collected either from the original literature or cited from the global database of Reich et al.[10]. In total, 103 publications (listed in Supplementary Text 2) with 382 and 1752 observations of $\delta^{15}N_{stem}$ and $\delta^{15}N_{root}$ were available, respectively (Fig. s2b, c). Sites of $\delta^{15}N_{stem}$ and $\delta^{15}N_{root}$ observations distribute between 45.8°S and 74.5°N (Fig. s1b), with the MAT spanning from −16.3 °C to 28.0 °C. The available $\delta^{15}N_{stem}$ and $\delta^{15}N_{root}$ observations were conducted for a total of 246 plant species, including 120 herbs, 57 shrubs, and 69 trees.

The $\delta^{15}N$ values showed difference between leaf and stem or root of the same individuals among the observed plants (Fig. s2d-f). The $\delta^{15}N_{leaf}$ and $\delta^{15}N_{plant}$ were positively correlated for herbs $\delta^{15}N_{plant} = 0.77(\pm 0.08) \times \delta^{15}N_{leaf} - 0.13(\pm 0.19)$ ($R = 0.93$, $p < 0.01$), shrubs $\delta^{15}N_{plant} = 0.80(\pm 0.06) \times \delta^{15}N_{leaf} - 0.98(\pm 0.10)$ ($R = 0.94$, $p < 0.01$), and trees $\delta^{15}N_{plant} = 0.93(\pm 0.00) \times \delta^{15}N_{leaf} - 0.13(\pm 0.00)$ ($R = 0.98$, $p < 0.01$) (Fig. s3c). These relationships were used to calculate $\delta^{15}N_{plant}$ values for each $\delta^{15}N_{leaf}$ observation of each life form in the global dataset (Fig. 2b).

## Constraining terrestrial $\delta^{15}N_{PUN}$ signatures

The differences between $\delta^{15}N_{plant}$ and $\delta^{15}N_{PUN}$ exist ($\Delta_m$) for plants associated with mycorrhiza due to substantial N isotope effects caused by plant N acquisition via mycorrhizal associations[22,24]. Thus, we have Eq. (3) to calculate the $\delta^{15}N_{PUN}$.

$$\delta^{15}N_{PUN} = \delta^{15}N_{plant} + \Delta_m \qquad (3)$$

To constrain the $\Delta_m$ values, we collected the records of the mycorrhizal types of plants either from the original literature or referring to records in other publications[40]. In this study, a total of 240, 4849, 391, and 106 among 5586 plant species are associated with NM, AM, ECM, and ERM, respectively. The $\Delta_m$ value for NM plant species was assumed as 0‰ because the direct entering processes of soil N sources into plant roots have no substantial $^{15}N$ discrimination[18,23,24]. The $\Delta_m$ values for plant species associated with mycorrhiza can be estimated as the $\delta^{15}N_{plant}$ differences between mycorrhizal plants and NM plants, respectively, which have been revealed varying in a general order of AM < ECM < ERM[24].

In this study, we newly found that the $\delta^{15}N_{plant}$ variations were influenced by life forms and MAT (Table s3, Fig. s4−5), thus we constructed the $\delta^{15}N_{plant}$ variations of each mycorrhizal type of each life form with MAT (Fig. s5). Then, we calculated the MAT-specific $\Delta_m$ values by using the fitting formula of NM minus the fitting formula of the corresponding each mycorrhizal type plants of the same life form (Table s4, Fig. s5). The negative correlation between MAT and $\Delta_m$ (Table s4) indicates that as MAT increases, soil N availability also increases, reducing plant reliance on mycorrhizae and thereby diminishing their N isotopic effects[9,24]. Therefore, we calibrated $\delta^{15}N_{plant}$ to $\delta^{15}N_{PUN}$ using this relationship to minimize the impact of mycorrhizae. Finally, the MAT-specific $\delta^{15}N_{PUN}$ values were calculated (Figs. s6a, 2c).

## Calculating contributions of soil $NO_3^-$, $NH_4^+$, and EON to PUN

The $\delta^{15}N_{PUN}$ value is determined by $\delta^{15}N$ values of soil $NO_3^-$, $NH_4^+$, and EON ($\delta^{15}N_{soil-NO3-}$, $\delta^{15}N_{soil-NH4+}$, and $\delta^{15}N_{soil-EON}$, respectively) and fractional contributions of soil $NO_3^-$, $NH_4^+$, and EON to PUN ($f_{NO3-}$, $f_{NH4+}$, and $f_{EON}$, respectively)[18,19,23], which can be expressed by Eq. (4).

$$\delta^{15}N_{PUN} = \delta^{15}N_{soil-NO3-} \times f_{NO3-} + \delta^{15}N_{soil-NH4+} \times f_{NH4+} \\ + \delta^{15}N_{soil-EON} \times f_{EON} \qquad (4)$$

where we assume that $f_{EON} + f_{NH4+} + f_{NO3-} = 1$.

To constrain the $\delta^{15}N_{soil-NO3-}$, $\delta^{15}N_{soil-NH4+}$, and $\delta^{15}N_{soil-EON}$ in terrestrial ecosystems, we searched literature published on the Web of Science and Google Scholar with the terms "nitrogen isotope", "$^{15}N$", "soil ammonium", "soil $NH_4^+$", "soil nitrate", "soil $NO_3^-$", "soil DON", "soil dissolved organic nitrogen", "soil EON", "soil extractable organic nitrogen". Observations for fertilized soils (including $^{15}N$-labeling experiments) and soils at depths over 30 cm were excluded from the searched literature. By August 1st, 2022, we obtained 46 publications (listed in Supplementary Text 3) with a total of 321, 356, and 262 observations of $\delta^{15}N_{soil-NO3-}$ at 56 sites, $\delta^{15}N_{soil-NH4+}$ at 62 sites, and $\delta^{15}N_{soil-EON}$ at 28 sites, respectively (Figs. s1c, s6b-d). Sites of soil $\delta^{15}N$

observations distribute between 30.3°S and 68.4°N, with the MAT spanning from −8.4 °C to 28.0 °C (Figs. s1c, 2d−f). To determine the regulating factor of soil $\delta^{15}N$ variations, we collected the data on soil density, clay, pH, and organic C from the Harmonized World Soil Database v1.2 (https://www.fao.org/home/en/) using the coordinate information. Besides, soil $NO_3^-$, $NH_4^+$, EON, and TEN concentrations were also collected for sites with simultaneous observations on $\delta^{15}N_{leaf}$, $\delta^{15}N_{soil-NO3-}$, $\delta^{15}N_{soil-NH4+}$, and $\delta^{15}N_{soil-EON}$ (Figs. s8, s11−12).

The $\delta^{15}N_{soil-NO3-}$, $\delta^{15}N_{soil-NH4+}$ and $\delta^{15}N_{soil-EON}$ values are influenced by various production and consumption processes of the corresponding N forms[17]. These processes, and thus the above soil $\delta^{15}N$ parameters, are influenced by climatic and environmental factors[37]. This study found that variations of $\delta^{15}N_{soil-NO3-}$, $\delta^{15}N_{soil-NH4+}$, and $\delta^{15}N_{soil-EON}$ were all influenced significantly by MAT (Table s5), increasing linearly with MAT (Fig. 2d−f). This finding is supported by evidence that increasing temperature enhances microbial N processes and the 'openness' of soil N cycles, leading to more $^{15}N$ enrichment in soil N sources[22,25,35]. Accordingly, we calculated the mean $\delta^{15}N_{soil-NO3-}$, $\delta^{15}N_{soil-NH4+}$, and $\delta^{15}N_{soil-EON}$ values for each $\delta^{15}N_{PUN}$ observation site by using the corresponding MAT and the fitting formula in Fig. 2d−f.

Then, we calculated $f_{NO3-}$, $f_{NH4+}$, and $f_{EON}$ values using an isotope-mixing model (named Stable Isotope Analysis in R, SIAR). The SIAR model, which is designed around a Bayesian framework, effectively utilizes a Dirichlet distribution to establish a logical prior for estimating source contributions[41,42]. This framework not only focuses on managing sample size variability but also prioritizes the data distribution within each sample over the mere number of observations, enhancing the model's accuracy and robustness in source estimation[41,42]. In each run of the SIAR model, the $\delta^{15}N_{PUN}$ data in a grid cell of 0.1° (latitude) × 0.1° (longitude) in the same sampling year (with 3−1028 observations for each grid cell in each year, referring to Craine et al.[26]), the mean ± SD of site-based $\delta^{15}N_{soil-NO3-}$, $\delta^{15}N_{soil-NH4+}$, and $\delta^{15}N_{soil-EON}$ values in the corresponding grid cell were input into the model. Then, the percentage data of each source ($n = 10,000$) output from each run of the SIAR model was used to calculate the mean ± SD of grid-based $f_{NO3-}$, $f_{NH4+}$, and $f_{EON}$ values (Fig. s7) and to map their spatial distribution (Fig. 3).

For study sites with simultaneous observations on $\delta^{15}N_{leaf}$, $\delta^{15}N_{soil-NO3-}$, $\delta^{15}N_{soil-NH4+}$, $\delta^{15}N_{soil-EON}$, and $\delta^{15}N_{soil-TEN}$, we separately calculated the site-based $\delta^{15}N_{plant}$, $\delta^{15}N_{PUN}$, $f_{NO3-}$, $f_{NH4+}$, and $f_{EON}$ values based on the above methods and examined their variations with MAT (Figs. s8, s11−12). In combination with simultaneous observations on soil $NO_3^-$, $NH_4^+$, EON, and TEN concentrations at each of the above study sites, we further calculated the relative preference degree ($\beta$) of plant N utilization among $NO_3^-$, $NH_4^+$, and EON by using Eq. (5).

$$\beta_i = f_i - ([i]/[TEN]) \qquad (5)$$

where i represents $NO_3^-$, $NH_4^+$, or EON, [i] and [TEN] are N concentrations of i and TEN, respectively. Positive, zero, and negative values of $\beta$ parameters indicated the preference of a given N form over other N forms, no preference, and the preference of the other N forms over a given N form, respectively[20,43].

## Statistical analyses

General Linear Models were used to examine the effects of major environmental and plant variables on $\delta^{15}N_{plant}$ variations of NM, AM, ECM and ERM plants. Multiple linear regression analyses were used to examine the effects of the major soil and environmental variables on soil $\delta^{15}N$ variations and plant N sources contribution. Regression analyses were used to examine variations of plant and soil $\delta^{15}N$ parameters and fractional contributions of soil N sources to PUN and TEN with MAT and latitude, respectively. The ArcGIS 10.8 software (Esri Inc., USA) was used to perform spatial interpolations, layers overlay and property sheet processing of data points. Regression analyses were

conducted using SPSS 16.0 for Windows (SPSS Inc., Chicago, IL, USA). Statistical significance was set at $p < 0.05$.

## Data availability
The data underlying the findings of this study are available in this article. Source data are provided with this paper.

## Code availability
The SPSS package can be downloaded from https://www.ibm.com/products/spss-statistics. The source code for SIAR used in this paper is openly available from https://rdrr.io/cran/siar/.

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

## Acknowledgements

This study was supported by the National Natural Science Foundation of China (42125301 (X.Y.L.), 42330505 (X.Y.L.), and 42103075 (C.C.H.)) and the Double Thousand Plan of Jiangxi Province (jxsq2023102213) (X.Y.L.). We want to thank all researchers who reported and kindly provided us with observation data on concentrations and isotopes of N in plants and soils.

## Author contributions

X.Y.L. designed the research. C.C.H. and X.Y.L. conducted the research (data collections and analyses) and co-wrote the manuscript. W.S., X.T.L., Y.W.K., C.J.C., R.M., E.N.J.B., A.W.D., B.Z.H., and C.Q.L. commented on the manuscript.

## Competing interests

The authors declare no competing interests.
