## [Peer Review File · Nature Communications]

Global distribution and drivers of relative contributions among soil nitrogen sources to terrestrial plantsREVIEWER COMMENTS

Reviewer #1 (Remarks to the Author):

This paper presents interesting research investigating the nitrogen isotope patterns within plant and soil systems. The authors have developed an impressive and updated global dataset of $\delta^{15}\text{N}$ of plants and soil, utilizing this dataset to explore the determining factors of nitrogen use patterns. Overall, this study is significant, the logic is clear, the analysis is robust, and the writing is articulate. However, I would like to highlight several major concerns regarding the isotope dataset and related calculations that need to be adequately addressed before considering publication.

1. There appears to be a cyclic logic in the calculations. The authors used MAT to correct Δm , $^{15}\text{N}_{\text{soil-NO}_3^-}$, $\delta^{15}\text{N}_{\text{soil-NH}_4^+}$, and $\delta^{15}\text{N}_{\text{soil-EON}}$. Later, these MAT-corrected values were employed to assess the relationship between nitrogen use and MAT. I am curious about how these MAT corrections may have influenced this relationship. This aspect is crucial for drawing conclusions.
2. In determining $\delta^{15}\text{N}_{\text{plant}}$, the authors employed a correlation equation to interpolate all $\delta^{15}\text{N}_{\text{leaf}}$ to $\delta^{15}\text{N}_{\text{plant}}$ on a global scale. Could the authors investigate whether there are functional group differences in the $\delta^{15}\text{N}_{\text{plant}}$ - $\delta^{15}\text{N}_{\text{leaf}}$ relationship? This could impact the generalizability of the correlation equation utilized here.
3. When calculating the fraction contributions from different nitrogen sources, the authors utilized SIAR model and mentioned the $\delta^{15}\text{N}_{\text{PUN}}$ data in a grid of 0.1° (latitude) \times 0.1° (longitude) in the same sampling year. Each grid had observations ranging from 3 to 1028 for each year. I am concerned about how such a wide range of sample sizes (with a three-magnitude difference) within each grid may impact the reliability of the results.

Some other specific comments:

1. The author mentioned that "However, one methodological question and one mechanistic question have not been resolved and answered in the quantification of soil N sources to PUN on the global scale". To me, both appear to be mechanistic questions.
2. The authors employed the SIAR model to calculate the contributions. While there are other isotope partitioning methods available, such as MixSIAR, which seem to be more commonly used, could the authors justify their choice of SIAR?

Reviewer #2 (Remarks to the Author):

This is hitherto a newest and most systematical analysis on global ^{15}N isotope data of plants (including leaves, stems, roots) and soils (ammonium, nitrate, EON, TEN) in land ecosystems. The authors designed a new regime to interpret leaf $\delta^{15}\text{N}$ signals and they proposed a new term 'PUN, i.e., plant-used N', which should be a new start to deeply understand the real plant N availability. This is really insightful and differs from existing studies on plant ^{15}N analyses. Reading the manuscript carefully and based on their descriptions, I found this work did contain some cutting-edge methods and new mechanism

findings.

In terms of research methods, it seems that no study has established the ^{15}N isotope relationships among organs measured for the same plant individuals, so that no study has tried to integrate the whole-plant N isotope signatures. Second, this seems also the first study examining and establishing the relationship between surface temperature and ^{15}N isotopic fractionations of mycorrhizal N uptake, without this, it is impossible to accurately calculate the $\delta^{15}\text{N}$ of PUN in soil. Those variation curves are valuable to re-check the dependence of plant N on mycorrhizal uptake pathways, I think. More, the authors moved a big step to estimate the N isotope variations of extractable N species in soils with MAT. I understand the data limitation of soil N sources for many sites and ecosystems and they are lucky to have the linear relationships for three N forms. Generally, I think these three steps did promote the current leaf ^{15}N from a qualitative index to be a quantitative tool for contributions of soil extractable N sources to the PUN.

From the research question and finding, as we knew, there are several works discussing progressive nitrogen limitation or declined N availability under increasing CO_2 based on decreasing leaf and tree ring $\delta^{15}\text{N}$. This paper seems providing solid evidence on the increasing availability and uptake of inorganic N particularly nitrate (^{15}N depleted) for decreasing or lower leaf $\delta^{15}\text{N}$! Anyway, I agree that plant N uptake manner, not N availability, appear a major control on leaf N isotope. It is also interesting to see that the temperature not N deposition and MAP control the general ^{15}N pattern of terrestrial ecosystems. According, we should rethink the stimulating effect of elevated N deposition on plant soil N cycles and reexamine the mechanism how increasing CO_2 and global warming can cause the PNL or declined N availability.

Totally, I think this study is timely and significant, focuses on an important topic, the method is effective and has several new improvements on natural isotope methods, their finding expands the understanding of ecosystem N dynamics. The manuscript is generally well organized and clearly written. However, the authors must address/explain the following issues before consideration of this manuscript.

1. Plant and soil N isotopes are widely recognized as effective indicators of ecosystem N cycling status and plant N sources. However, practical applications still face unresolved issues. Despite this manuscript's thorough consideration of fractionation effects in plant N utilization processes compared to previous studies, uncertainties persist in all research endeavors. Therefore, I recommend the authors supplement the discussion with a section addressing uncertainty. Furthermore, I encourage the inclusion of insights into future research prospects, such as which aspects to prioritize given these uncertainties, and how the temperature-regulated changes in plant nitrogen utilization revealed by the study could impact ecosystem carbon dynamics in the future.
2. The manuscript presents a highly intriguing conclusion suggesting that atmospheric N deposition does not significantly impact plant N utilization, contrary to findings from some N addition experiments. I find this conclusion entirely plausible and of significant importance. Several factors likely contribute to the observed lack of impact of in-situ N utilization by plants due to atmospheric N deposition. Firstly, it is important to note that not all regions globally experience high levels of N deposition; indeed, the proportion of areas with high N deposition remains relatively low on a global scale. Secondly, the amount of N deposited from the atmosphere is minuscule compared to the total pool of plant-available N in the soil. Thirdly, differences between the results of N addition experiments and those observed in-situ under natural conditions may arise from the controlled nature of experiments, where only N addition is considered as a single influencing factor, while natural conditions are inherently

complex. I believe it would greatly enhance the manuscript for the authors to include these insightful discussions in the Discourse section.

3. The relationship plots of f values varying with latitude in Figures 3a-c serve as initial data for kriging interpolation. Although this initial data already effectively reflects the variation pattern of plant nitrogen utilization with latitude, I suggest placing these relationship plots in the supporting materials. Instead, this section should feature plots showing the fitting of global kriging interpolation results obtained with latitude.

Moreover, the paper should be checked following minor points to improve the clarity and readability.

Line 24: Replace "nutrients" with "sources".

Line 26: " $\delta^{15}\text{N}$ " can be removed.

Line 45-26: Replace "resources" with "sources".

Line 48: Because "carbon" appears infrequently, please delete "(C)" and simply use "carbon".

Line 52: Replace "nutrients" with "sources".

Line 62: "chemical" can be removed.

Line 75-78: The evidence supporting this statement can be found in Table 1 of the authors' supporting materials. Please cited Table in here.

Line 106: Replace "resources" with "sources".

Line 109: Replace "mean annual temperature (MAT)" with "MAT".

Line 118: Replace "a globally open question" with "an open question at global scale".

Line 133: Replace "experimental duration" with "and experimental duration".

Line 141: The order of the supplementary materials in the supporting information should be adjusted so that "Supplementary Text 3" becomes "Supplementary Text 1".

Line 155: Replace " $0.1 \times 0.1^\circ$ " with " 0.1° (latitude) \times 0.1° (longitude)".

Line 157: Replace "relative" with "fractional".

Line 192-194: I believe this sentence should be replaced with the results of plant nitrogen utilization varying with latitude.

Line 243: Replace "soil PUN" with "PUN".

Line 247: Replace "plant-used N" with "PUN".

Line 285: I believe the mean values of $\Delta f\text{NO}_3^-$, $\Delta f\text{NH}_4^+$, and $\Delta f\text{EON}$ should be explicitly stated.

Line 287: In sites where N concentrations and isotopes are observed simultaneously, although there is a consistent trend in plant N utilization compared to the global scale, it may not be statistically significant due to the limited number of sites and the presence of considerable errors. Therefore, the insignificance of the results should also be explicitly stated here.

Line 289-290: The sentence should express that the detailed calculation method of β has been presented in the Methods section.

Line 308: "co-existence and invasion, diversity and succession" can be removed.

Line 312: The term "future environmental changes" encompasses the meaning of "climate change," therefore, it can be modified as "forthcoming environmental shifts".

Line 449-450: 5632 do not equal to the sum of 2170, 1146 and 2299.

Line 449: Replace "species" with "plant species".

Line 463-464: To avoid ambiguity, the sentence should be rewritten.

Line 481-483: Replace "While the $\delta^{15}\text{N}_{\text{leaf}}$ values did not differ substantially from the corresponding $\delta^{15}\text{N}_{\text{plant}}$ values, showing a positive correlation of" with "The $\delta^{15}\text{N}_{\text{leaf}}$ and $\delta^{15}\text{N}_{\text{plant}}$ values showing a positive correlation of".

Line 561: "(GLMs)" can be removed.

In the supporting materials, Figures 10a-c should include fitted lines. Although not significant, the trends are consistent with those at the global scale, so adding trend lines would enhance clarity.

Reviewer #3 (Remarks to the Author):

The manuscript titled "Surface temperature regulated global nitrogen-use patterns of terrestrial plants" by Hu et al. presents, for the first time, simultaneous reports of N isotope data from global leaf, root, stem, and soil N in various forms. It employs innovative methodologies to analyze global patterns of plant N utilization. This work is timely and holds significant importance in advancing our comprehension of global-scale patterns of plant-soil N isotope dynamics and their reflection on terrestrial ecosystem N cycling and global plant N utilization mechanisms. For instance, the manuscript offers a comprehensive overview of previous research methodologies, upon which it constructs a novel isotope geochemistry model, thereby significantly advancing the application of N isotopes in ecosystem N cycling. Simultaneously, it provides a thorough review of current progress in plant N utilization. Building upon this foundation, the key findings of this manuscript renew our understanding of global plant N utilization and its regulatory mechanisms compared to previous research. The research methods, content, and key findings of this article are likely to arouse the interest of readers from various research backgrounds such as isotope geochemistry, global change, environmental science, ecology, and plant physiological ecology. Additionally, the manuscript is generally adequately presented and clearly structured. Therefore, I suggested accepting this manuscript for publication in Nature Communications after minor revisions.

1. The manuscript asserts that the findings presented in this study significantly contribute to our understanding of alterations in ecosystem structure and functions, spanning productivity, carbon sequestration, species coexistence, exotic plant invasion, and diversity (lines 305-308). However, as noted by the authors in the Introduction, a precise understanding of PUN is crucial for assessing terrestrial ecosystem carbon cycling. Therefore, it is imperative for the authors to emphasize the implications of global plant N utilization patterns and their variations on ecosystem carbon cycling.

2. Plants incur lower energy expenditures during the assimilation of ammonium in comparison to nitrate, prompting certain plants to preferentially utilize ammonium. However, an excess of ammonium can prove deleterious to plants. The study identifies a gradual decline in the contribution of ammonium post a particular threshold. Apart from the rationales delineated by the authors, such as potential enhancements in nitrification (line 280), it is imperative to acknowledge the toxic repercussions of ammonium. Thus, I propose the inclusion of this aspect within the discussion.

Minor comments:

Line 26: I suggested to remove ' $\delta^{15}\text{N}$ ' as it appears only once in the Abstract, and it is also explained in the Introduction section later.

Line 64: C/N is not only used to model N cycling but are also commonly employed in C cycling simulations.

Line 71: 'Chemical structure' should be changed to 'chemical proportions'

Lines 108-112: This sentence is too long; it is advisable to split it into two sentences.

Line 136: 'Here' is simpler than 'Based on the above context'

Lines 165, 180, 191: Since it is already understood that the data integration is global in scope, I suggest removing 'Globally' from these three places.

Line 167: I propose removing 'on average'.

Lines 192-194: I suggest deleting or modifying this sentence to reflect the variation of N

utilization with latitude.

Line 230: Change 'On the global scale' to 'Globally'

Lines 264-266: There is an expression error in this sentence that needs to be corrected

Lines 514-517: The database source information for soil data should be provided.

Lines 541, 542, 544: Change 'grid' to 'grid cell'.

Authors' responses to review comments

Nature Communications (NCOMMS-24-12058-T)

Levels and spatial variations of soil nitrogen source contributions to global terrestrial plants

Reviewer #1 (Remarks to the Author):

This paper presents interesting research investigating the nitrogen isotope patterns within plant and soil systems. The authors have developed an impressive and updated global dataset of $\delta^{15}\text{N}$ of plants and soil, utilizing this dataset to explore the determining factors of nitrogen use patterns. Overall, this study is significant, the logic is clear, the analysis is robust, and the writing is articulate. However, I would like to highlight several major concerns regarding the isotope dataset and related calculations that need to be adequately addressed before considering publication.

Thank you for reviewing our manuscript and constructive comments.

1. There appears to be a cyclic logic in the calculations. The authors used MAT to correct Δ_m , $\delta^{15}\text{N}_{\text{soil-NO}_3^-}$, $\delta^{15}\text{N}_{\text{soil-NH}_4^+}$, and $\delta^{15}\text{N}_{\text{soil-EON}}$. Later, these MAT-corrected values were employed to assess the relationship between nitrogen use and MAT. I am curious about how these MAT corrections may have influenced this relationship. This aspect is crucial for drawing conclusions.

Response: Thank you for very insightful comments. Circular logic exists when using conclusions as premises in the same argument, which did not exist in our calculations.

There were two major groups of parameters (soil N isotopes, isotope effects of Δ_m values) which were estimated or calibrated by using their relationships with the same factor MAT. These relationships were not really or purely empirical, but driven by observation data and logically intrinsic. The calculated fractional contributions were integrative expressions of plant-soil isotopes and isotope effects, their variation relationships with MAT might be influenced or transferred from the above parameters, but they are not taken as premises in our calculations. Our calculations or isotopic interpretations were not circular but a unidirectional deciphering and logically supportive by N-cycle theories.

In our study, the regression analyses identified that MAT (not other factors) was the main regulator on the variations of plant-soil N isotopes (Table s5) or isotopic effects (Table s3-4), i.e., their relationships with MAT were established by existing observation data. The positive correlations between soil N isotopes and MAT are supported by well-documented theories of soil N cycles that warming climates stimulate soil N-cycle openness and ^{14}N losses (Amundson et al., 2003; Craine et al., 2015). Meanwhile, stimulated N mineralization and nitrification under warmer climates caused higher amounts and direct uptake fractions (relative to the mycorrhizal pathways) of bioavailable N in soils (Hobbie & Högberg, 2012). The negative correlations between Δ_m and MAT were first found being generally consistent for plant functional groups (Table s4), which pointed to the fact that rising temperatures increase soil N availability and reduce plants' reliance on mycorrhizae

for N acquisition, diminishing the overall isotopic effects of plant N uptake.

However, for sites without soil N isotope observations or with observations in different sampling time, we did estimate soil N source isotopes based on their relationships with MAT. Similarly, plant Δ_m values of sites with no leaf observations (Fig. s5) were also extrapolated by using their relationships with MAT (Table s4). For those sites, there were substantial uncertainties in calculated fractions of soil N source contributions and their variation patterns with MAT for those sites. Fortunately, for the majority of sites supporting our analysis and main conclusion, their variations with MAT were identified by reasonable methods and calculated results for these sites have been justified by using the simultaneous observation data (Figs. s7, s11).

Thanks to your insightful and helpful comments, we thought it is necessary to fully and clearly explain these logics and uncertainties in our manuscript because they may cause a misunderstanding of our unidirectional deciphering and calculations as circular logic. Please check our revisions in lines 279-281, 308-319, 515-520, and 549-551.

2. In determining $\delta^{15}\text{N}_{\text{plant}}$, the authors employed a correlation equation to interpolate all $\delta^{15}\text{N}_{\text{leaf}}$ to $\delta^{15}\text{N}_{\text{plant}}$ on a global scale. Could the authors investigate whether there are functional group differences in the $\delta^{15}\text{N}_{\text{plant}}-\delta^{15}\text{N}_{\text{leaf}}$ relationship? This could impact the generalizability of the correlation equation utilized here.

Response: This is very helpful suggestion. We examined differences in the $\delta^{15}\text{N}_{\text{leaf}}-\delta^{15}\text{N}_{\text{plant}}$ relationship among plant functional groups and did identify differences among herb, shrub, and tree (detailed in Fig. s3c). Accordingly, we recalibrated the relationship for calculating $\delta^{15}\text{N}_{\text{plant}}$ values of each functional group and therefore updated downstream calculations of Δ_m and $\delta^{15}\text{N}_{\text{PUN}}$. These modifications did not change the patterns of soil N source contributions and the main conclusion of our study, but make our calculations more accurate.

Thank you so much! We have explained the calibrations in lines 490-495 of Methods and updated the corresponding results in Figs. 2-4, s3-6, s8-11, Tables s3-4, and lines 30-35, 191-197.

3. When calculating the fraction contributions from different nitrogen sources, the authors utilized SIAR model and mentioned the $\delta^{15}\text{N}_{\text{PUN}}$ data in a grid of 0.1° (latitude) \times 0.1° (longitude) in the same sampling year. Each grid had observations ranging from 3 to 1028 for each year. I am concerned about how such a wide range of sample sizes (with a three-magnitude difference) within each grid may impact the reliability of the results.

Response: Thank you for pointing this out. Observation sizes did differ greatly among grids. Ideally, the more observations there are in each grid, the more representative the final results would be. However, we utilized the SIAR model, which is specifically crafted to manage observation sizes variability efficiently through its Bayesian framework. This framework prioritizes data distribution within each observation over observation sizes, seamlessly integrating uncertainty into the model's calculations. Thus, this method not only addresses sample size variability inherently but also enhances the flexibility and accuracy of estimating source contributions, especially valuable when observation sizes are inconsistent across

different groups or conditions. Based on your comments, we analyzed the variability of (CV: ratio of standard deviation to mean) $\delta^{15}\text{N}_{\text{PUN}}$, $f_{\text{NO}_3^-}$, $f_{\text{NH}_4^+}$, and f_{EON} across grids and found no clear relationship with observation sizes (see the following attached figure). Accordingly, these results suggest that the high variability of observation sizes might not substantially bias our estimates and the global patterns of soil N source contributions.

Based on the above content, we have provided further explanations in the corresponding sections of the manuscript. Please check lines 555-560.

Some other specific comments:

1. The author mentioned that “However, one methodological question and one mechanistic question have not been resolved and answered in the quantification of soil N sources to PUN on the global scale”. To me, both appear to be mechanistic questions.

Response: Thank you for your insights. Addressing the N isotope fractionation effects of intra-plant N assimilation/allocation and mycorrhizal N uptake processes is indeed a fundamental mechanistic question. It is critical for understanding ecosystem N cycling and plant N utilization, as reflected by $\delta^{15}\text{N}_{\text{leaf}}$ values. Thus, we have revised this sentence to "However, two fundamental questions concerning the quantification of soil N sources to PUN remain unresolved at the global scale" (please check lines 92-93), and also changed "Methodologically" to "First" (please check line 93) and changed "Mechanistically" to "Second" (please check line 105).

2. The authors employed the SIAR model to calculate the contributions. While there are other isotope partitioning methods available, such as MixSIAR, which seem to be more commonly used, could the authors justify their choice of SIAR?

Response: Thanks. Yes. MixSIAR, introduced by Stock et al (2016), represents the latest Bayesian mixing model. Compared to SIAR, MixSIAR incorporates various source data input forms, fixed effects, and random effects. Essentially, when multiple source input forms, fixed effects, and random effect parameters are present, MixSIAR can enhance the computational precision of the model. However, this study does not require consideration of parameters such as fixed effects and random effects, thus utilizing SIAR completely fulfills the requirements (please check lines 555-560).

Reviewer #2 (Remarks to the Author):

This is hitherto a newest and most systematical analysis on global ^{15}N isotope data of plants (including leaves, stems, roots) and soils (ammonium, nitrate, EON, TEN) in land ecosystems. The authors designed a new regime to interpret leaf $\delta^{15}\text{N}$ signals and they proposed a new term 'PUN, i.e., plant-used N', which should be a new start to deeply understand the real plant N availability. This is really insightful and differs from existing studies on plant ^{15}N analyses. Reading the manuscript carefully and based on their descriptions, I found this work did contain some cutting-edge methods and new mechanism findings.

In terms of research methods, it seems that no study has established the ^{15}N isotope relationships among organs measured for the same plant individuals, so that no study has tried to integrate the whole-plant N isotope signatures. Second, this seems also the first study examining and establishing the relationship between surface temperature and ^{15}N isotopic fractionations of mycorrhizal N uptake, without this, it is impossible to accurately calculate the $\delta^{15}\text{N}$ of PUN in soil. Those variation curves are valuable to re-check the dependence of plant N on mycorrhizal uptake pathways, I think. More, the authors moved a big step to estimate the N isotope variations of extractable N species in soils with MAT. I understand the data limitation of soil N sources for many sites and ecosystems and they are lucky to have the linear relationships for three N forms. Generally, I think these three steps did promote the current leaf ^{15}N from a qualitative index to be a quantitative tool for contributions of soil extractable N sources to the PUN.

From the research question and finding, as we knew, there are several works discussing progressive nitrogen limitation or declined N availability under increasing CO_2 based on decreasing leaf and tree ring $\delta^{15}\text{N}$. This paper seems providing solid evidence on the increasing availability and uptake of inorganic N particularly nitrate (^{15}N depleted) for decreasing or lower leaf $\delta^{15}\text{N}$! Anyway, I agree that plant N uptake manner, not N availability, appear a major control on leaf N isotope. It is also interesting to see that the temperature not N deposition and MAP control the general ^{15}N pattern of terrestrial ecosystems. According, we should rethink the stimulating effect of elevated N deposition on plant soil N cycles and reexamine the mechanism how increasing CO_2 and global warming can cause the PNL or declined N availability.

Totally, I think this study is timely and significant, focuses on an important topic, the method is effective and has several new improvements on natural isotope methods, their finding expands the understanding of ecosystem N dynamics. The manuscript is generally well organized and clearly written. However, the authors must address/explain the following issues before consideration of this manuscript.

Thank you for reviewing our manuscript.

1. Plant and soil N isotopes are widely recognized as effective indicators of ecosystem N cycling status and plant N sources. However, practical applications still face unresolved issues. Despite this manuscript's thorough consideration of fractionation effects in plant N utilization processes compared to previous studies, uncertainties persist in all research endeavors. Therefore, I recommend the authors supplement the discussion with a section addressing uncertainty. Furthermore, I encourage the inclusion of insights into future research prospects, such as which aspects to prioritize given these uncertainties, and how the temperature-regulated changes in plant nitrogen utilization revealed by the study could impact ecosystem carbon dynamics in the future.

Response: Thank you for your insightful suggestions, which will significantly aid our future work. In this study, although we considered the isotopic effects in plant N utilization processes and constructed global soil N source isotope values based on all available plant and soil N isotope data, some uncertainties still remain. Extant $\delta^{15}\text{N}$ observations on roots and stems were still sparse among terrestrial plants (Fig. s2), simultaneous observations on $\delta^{15}\text{N}$, N concentrations, and biomass for leaves, stems, and roots were even less (Fig. s3c). Particularly, $\delta^{15}\text{N}$ values of soil extractable N sources have seldom been observed together with those of plants, particularly for sites and ecosystems with relatively lower MAT (Figs. 2, s6-7). More, $\delta^{15}\text{N}$ values of soil extractable organic N sources that were really used by plants remain difficult to be determined. These are substantial uncertainties in the established relationships and calculated results of this study. Although it might not influence the general patterns of soil N source contributions to global plants found in the present study, to widely conduct simultaneous and point-to-point observations on plant-soil $\delta^{15}\text{N}$ parameters would provide more precise estimates on plant N sources and availability for biogeochemical and earth system modeling efforts.

Considering the coupled relationship between carbon and N in ecosystems, temperature-regulated plant N utilization will affect ecosystem carbon (C) dynamics in the following ways. Because of differing C costs among plant NH_4^+ , NO_3^- , EON assimilations, the temperature regulation of plant utilization of these N sources suggests that warming climate would change the overall C costs and influences the ability of ecosystems to fixing CO_2 in the atmosphere. These dynamics are crucial for predicting how terrestrial C stocks will respond to global warming.

Following your suggestions, we have revised the section of Concluding remarks and uncertainties (please check line 291-319).

2. The manuscript presents a highly intriguing conclusion suggesting that atmospheric N deposition does not significantly impact plant N utilization, contrary to findings from some N addition experiments. I find this conclusion entirely plausible and of significant importance. Several factors likely contribute to the observed lack of impact of in-situ N utilization by plants due to atmospheric N deposition. Firstly, it is important to note that not all regions globally experience high levels of N deposition; indeed, the proportion of areas with high N deposition remains relatively low on a global scale. Secondly, the amount of N deposited from the atmosphere is minuscule compared to the total pool of plant-available N in the soil. Thirdly, differences between the results of N addition experiments and those observed in-situ under

natural conditions may arise from the controlled nature of experiments, where only N addition is considered as a single influencing factor, while natural conditions are inherently complex. I believe it would greatly enhance the manuscript for the authors to include these insightful discussions in the Discourse section.

Response: Thank you for your comments and for recognizing the significance of our study's conclusions. We fully agree with your perspective that the discrepancies between our findings and those from N addition experiments simulating N deposition can be attributed to three primary reasons. However, because our study does not have substantial evidence to support these three aspects you mentioned, we cannot include them in the discussion without risking overinterpretation. In fact, as shown in Supplementary Tables s2, even N addition experiments can yield different results due to factors such as the duration and amount of N added. As mentioned in the Introduction (lines 118-134), plant N utilization under natural conditions is likely influenced by multiple factors, unlike controlled N addition experiments. Our study indicates that on a global scale, under natural conditions, N deposition and MAP do not significantly alter plant N utilization, whereas MAT plays a critical role in driving the geographical distribution of plant N-use patterns. Based on this information, we have added further explanations to the Discussion section. Please refer to lines 255–258.

3. The relationship plots of f values varying with latitude in Figures 3a-c serve as initial data for kriging interpolation. Although this initial data already effectively reflects the variation pattern of plant nitrogen utilization with latitude, I suggest placing these relationship plots in the supporting materials. Instead, this section should feature plots showing the fitting of global kriging interpolation results obtained with latitude.

Response: Thanks for your suggestion. Following your suggestion, we have updated Fig. 3 with new plots that depict the relationship between latitude and plant N utilization derived from kriging interpolation results (please check Fig. 3). The original plots have been moved to the supplementary materials (please check Fig. s9).

Moreover, the paper should be checked following minor points to improve the clarity and readability.

Line 24: Replace "nutrients" with "sources".

Line 26: " $\delta^{15}\text{N}$ " can be removed.

Line 45,46: Replace "resources" with "sources".

Line 48: Because "carbon" appears infrequently, please delete "(C)" and simply use "carbon".

Line 52: Replace "nutrients" with "sources".

Line 62: "chemical" can be removed.

Line 75-78: The evidence supporting this statement can be found in Table 1 of the authors' supporting materials. Please cited Table in here.

Line 106: Replace "resources" with "sources".

Line 109: Replace "mean annual temperature (MAT)" with "MAT".

Line 118: Replace "a globally open question" with "an open question at global scale".

Line 133: Replace "experimental duration" with "and experimental duration".

Line 155: Replace " $0.1 \times 0.1^\circ$ " with " 0.1° (latitude) \times 0.1° (longitude)".

Line 157: Replace "relative" with "fractional".
Line 243: Replace "soil PUN" with " PUN ".
Line 247: Replace " plant-used N " with " PUN ".
Line 308: " co-existence and invasion, diversity and succession " can be removed.
Line 312: The term "future environmental changes" encompasses the meaning of "climate change," therefore, it can be modified as "forthcoming environmental shifts".
Line 449: Replace " species " with " plant species ".
Line 481-483: Replace "While the $\delta^{15}\text{N}_{\text{leaf}}$ values did not differ substantially from the corresponding $\delta^{15}\text{N}_{\text{plant}}$ values, showing a positive correlation of " with "The $\delta^{15}\text{N}_{\text{leaf}}$ and $\delta^{15}\text{N}_{\text{plant}}$ values showing a positive correlation of ".
Line 561: "(GLMs)" can be removed.

Response: These have been corrected according to your comments and checked carefully through in the revised manuscript. Thanks a lot.

Line 141: The order of the supplementary materials in the supporting information should be adjusted so that "Supplementary Text 3" becomes "Supplementary Text 1".

Response: Thanks for pointing this out. We have rearranged the order of the Supplementary Text, shifting what was originally Supplementary Text 3 to the position of Supplementary Text 1. We have also made corresponding adjustments throughout the main text to reflect this new sequence.

Line 192-194: I believe this sentence should be replaced with the results of plant nitrogen utilization varying with latitude.

Response: Thanks for your suggestion. We have updated the sentence to reflect the relationship between plant N utilization and latitude (please check line 188-189).

“Generally, the $f_{\text{NO}_3^-}$ and $f_{\text{NH}_4^+}$ increased while the f_{DON} decreased with the latitude (Figs. 3,s9).”

Line 285: I believe the mean values of $\Delta f_{\text{NO}_3^-}$, $\Delta f_{\text{NH}_4^+}$, and Δf_{EON} should be explicitly stated.

Response: Thanks. Following your suggestion, we have separately presented the values for $\Delta f_{\text{NO}_3^-}$, $\Delta f_{\text{NH}_4^+}$, and Δf_{EON} (please check line 278-279).

Line 287: In sites where N concentrations and isotopes are observed simultaneously, although there is a consistent trend in plant N utilization compared to the global scale, it may not be statistically significant due to the limited number of sites and the presence of considerable errors. Therefore, the insignificance of the results should also be explicitly stated here.

Response: Thanks for pointing this out. Following your suggestion, we have added a statement regarding the non-significance in this sentence (please check line 281-282).

Line 289-290: The sentence should express that the detailed calculation method of β has been presented in the Methods section.

Response: Thanks for pointing this out. Following your suggestion, we have added

“detailed in Methods” in this sentence (please check line 284).

Line 449-450: 5632 do not equal to the sum of 2170, 1146 and 2299.

Response: Thanks for pointing this out. We conducted a detailed census of the plant functional groups here and proceeded with revisions (please check line 457-458).

Line 463-464: To avoid ambiguity, the sentence should be rewritten.

Response: Thanks. Based on your advice, we have revised this sentence (please check line 471-473).

“where N_{leaf} , N_{stem} , and N_{root} are the N concentrations in the leaf, stem, and root, respectively; F_{leaf} , F_{stem} , and F_{root} are their respective biomass percentages in the whole plant.”

In the supporting materials. Figure s10a-c should include fitted lines. Although not significant, the trends are consistent with those at the global scale, so adding trend lines would enhance clarity.

Response: Thanks for your suggestion. Based on your advice, we have incorporated fitted lines and their confidence intervals into Fig. s11a-c (please check Fig. s11).

Reviewer #3 (Remarks to the Author):

The manuscript titled "Surface temperature regulated global nitrogen-use patterns of terrestrial plants" by Hu et al. presents, for the first time, simultaneous reports of N isotope data from global leaf, root, stem, and soil N in various forms. It employs innovative methodologies to analyze global patterns of plant N utilization. This work is timely and holds significant importance in advancing our comprehension of global-scale patterns of plant-soil N isotope dynamics and their reflection on terrestrial ecosystem N cycling and global plant N utilization mechanisms. For instance, the manuscript offers a comprehensive overview of previous research methodologies, upon which it constructs a novel isotope geochemistry model, thereby significantly advancing the application of N isotopes in ecosystem N cycling. Simultaneously, it provides a thorough review of current progress in plant N utilization. Building upon this foundation, the key findings of this manuscript renew our understanding of global plant N utilization and its regulatory mechanisms compared to previous research. The research methods, content, and key findings of this article are likely to arouse the interest of readers from various research backgrounds such as isotope geochemistry, global change, environmental science, ecology, and plant physiological ecology. Additionally, the manuscript is generally adequately presented and clearly structured. Therefore, I suggested accepting this manuscript for publication in Nature Communications after minor revisions.

Thank you for reviewing our manuscript.

1. The manuscript asserts that the findings presented in this study significantly

contribute to our understanding of alterations in ecosystem structure and functions, spanning productivity, carbon sequestration, species coexistence, exotic plant invasion, and diversity (lines 305-308). However, as noted by the authors in the Introduction, a precise understanding of PUN is crucial for assessing terrestrial ecosystem carbon cycling. Therefore, it is imperative for the authors to emphasize the implications of global plant N utilization patterns and their variations on ecosystem carbon cycling.

Response: Thanks for your constructive comment, which aligns with the first comment made by Reviewer #2. Our research has implications in two aspects. Firstly, this study presents a quantitative analysis on levels and spatial variations of soil N source contributions to global terrestrial plants, providing new methods and evidences for evaluating plant N-use mechanisms and N cycle of terrestrial ecosystems. By explicitly constraining isotope effects of intra-plant N assimilation/allocation and mycorrhizal N acquisition of different plant life forms under different climates, we first transferred the $\delta^{15}\text{N}_{\text{leaf}}$ to a new parameter of $\delta^{15}\text{N}_{\text{PUN}}$ for elucidating plant N utilization. Substantially differing signatures between $\delta^{15}\text{N}_{\text{PUN}}$ and $\delta^{15}\text{N}_{\text{TEN}}$ informed that neither $\delta^{15}\text{N}_{\text{leaf}}$ nor $\delta^{15}\text{N}_{\text{TEN}}$ can be directly taken as an indicator of soil N availability or the relative availability of PUN to soil N supply. Secondly, considering the coupled relationship between carbon and N in ecosystems, temperature-regulated plant N utilization will affect ecosystem carbon dynamics in the following ways. Because of differing C costs among plant NH_4^+ , NO_3^- , EON assimilations, the temperature regulation of plant utilization of these N sources suggests that warming climate would change the overall C costs and influences the ability of ecosystems to fixing CO_2 in the atmosphere. These dynamics are crucial for predicting how terrestrial C stocks will respond to global warming.

Following your suggestions, we have revised the section of Concluding remarks and uncertainties (please check line 291-307).

2. Plants incur lower energy expenditures during the assimilation of ammonium in comparison to nitrate, prompting certain plants to preferentially utilize ammonium. However, an excess of ammonium can prove deleterious to plants. The study identifies a gradual decline in the contribution of ammonium post a particular threshold. Apart from the rationales delineated by the authors, such as potential enhancements in nitrification (line 280), it is imperative to acknowledge the toxic repercussions of ammonium. Thus, I propose the inclusion of this aspect within the discussion.

Response: Thank you for your suggestion. In this study, we observed a decline in $f_{\text{NH}_4^+}$ after it reached 46%. Indeed, beyond the increased nitrification discussed in this study, the toxicological effects of ammonium on plants that you pointed out might also contribute. This is supported by findings that excessive ammonium uptake and assimilation can result in decreases in intracellular pH and ionic imbalance in plants (Britto & Kronzucker 2002). We have added more explanations to Discussion section. Please check lines 271-274.

Minor comments:

Line 26: I suggested to remove ' $\delta^{15}\text{N}$ ' as it appears only once in the Abstract, and it is

also explained in the Introduction section later.

Line 71: 'Chemical structure' should be changed to 'chemical proportions'

Line 136: 'Here' is simpler than 'Based on the above context'

Lines 165, 180, 191: Since it is already understood that the data integration is global in scope, I suggest removing 'Globally' from these three places.

Line 167: I propose removing 'on average'.

Line 230: Change 'On the global scale' to 'Globally'

Lines 541, 542, 544: Change 'grid' to 'grid cell'.

Response: These have been corrected according to your comments and checked carefully through in the revised manuscript. Thanks a lot.

Line 64: C/N is not only used to model N cycling but are also commonly employed in C cycling simulations.

Response: Yes, thank you for pointing this out. Based on your suggestion, we have also included the carbon cycle in this statement (please check line 64).

Lines 108-112: This sentence is too long; it is advisable to split it into two sentences.

Response: Thanks, we have split the sentence into two, based on your suggestion (please check line 107-112).

“Regarding the contributing proportions, some studies showed that plants under lower MAT (< 5°C) or at higher latitudes (> 63°N) mainly utilized EON (43–66% (c.a. >59%) for tundra plants). In contrast, the other studies in these regions estimated much lower contributions of organic N (c.a. < 22%), with c.a. 14–61% and 24–63% from NH₄⁺ and NO₃⁻, respectively (data compiled in Table S1).”

Lines 192-194: I suggest deleting or modifying this sentence to reflect the variation of N utilization with latitude.

Response: Thanks for your suggestion. We have updated the sentence to reflect the relationship between plant N utilization and latitude (please check line 188-189).

“Generally, the $f_{NO_3^-}$ and $f_{NH_4^+}$ increased while the f_{DON} decreased with the latitude (Figs. 3, S9).”

Lines 264-266: There is an expression error in this sentence that needs to be corrected

Response: Thanks, we have revised this sentence (please check line 258-260).

“From polar to tropical regions, both $f_{NO_3^-}$ and $f_{NH_4^+}$ increased, and f_{EON} decreased, with the decrease in latitude (Fig. 3, S9) and the increase in MAT (Fig. 4).”

Lines 514-517: The database source information for soil data should be provided.

Response: Thanks for pointing this out. The compilation of soil N isotope data was sourced from publications indexed on Web of Science and Google Scholar. We have

added this information to the sentence (please check line 529-530).

References:

- Amundson R, Austin AT, Schuur EAG, et al. 2003. Global patterns of the isotopic composition of soil and plant nitrogen. *Global Biogeochemical Cycles*, 17, 1031–1041.
- Britto DT, Kronzucker HJ. 2002. NH_4^+ toxicity in higher plants: a critical review. *Journal of Plant Physiology*, 159, 567–584.
- Craine JM, Brookshire ENJ, Cramer MD, et al. 2015. Ecological interpretations of nitrogen isotope ratios of terrestrial plants and soils. *Plant Soil*, 396, 1–26.
- Hobbie EA, Högberg P. 2012. Nitrogen isotopes link mycorrhizal fungi and plants to nitrogen dynamics. *New Phytologist*, 196, 376–382.
- Stock BC, Semmens BX. 2016. Unifying error structures in commonly used biotracer mixing models. *Ecology*, 97, 2562–2569.

REVIEWERS' COMMENTS

Reviewer #1 (Remarks to the Author):

I appreciate the authors' significant efforts to address the reviewers' comments. Most of my comments related to science have been addressed satisfactorily.

There are some additional comments to further improve the manuscript:

Abstract and Line 192, It is hard to imagine plant N uptake under “-18.7oC” temperature. This data point also seems to affect the overall relationships. Please double-check the cold temperature data points.

The current title is not very informative. I would suggest changing it to “A new framework to estimate soil nitrogen contributions to global terrestrial plants”, or “Novel insights of soil nitrogen contributions to global terrestrial plants”, or something similar.

Line 228 I don't think nitrate leaching has strong isotope fractionation. Please double-check this statement.

More importantly, the language of the manuscript could be further polished to enhance readability and the impact it deserves.

Minor issues:

Line 166 “Figs.” should be “Fig.”

Line 202 “Figs.” should be “Fig.”

“evidnces” in “providing new methods and evidences for evaluating plant N-use mechanisms and N cycle of terrestrial ecosystems.” should be “evidence”.

Reviewer #2 (Remarks to the Author):

The authors have well addressed my concerns and I suggest this manuscript for publication.

Reviewer #3 (Remarks to the Author):

The authors have well addressed my comments. I am happy to recomand acceptance.

Authors' responses to review comments

Nature Communications (NCOMMS-24-12058A)

Global distribution and drivers of relative contributions among soil nitrogen sources to terrestrial plants

Reviewer #1 (Remarks to the Author):

I appreciate the authors' significant efforts to address the reviewers' comments. Most of my comments related to science have been addressed satisfactorily.

Once again, we sincerely thank you for reviewing our manuscript and providing valuable comments.

There are some additional comments to further improve the manuscript:

Abstract and Line 192, It is hard to imagine plant N uptake under “-18.7°C” temperature. This data point also seems to affect the overall relationships. Please double-check the cold temperature data points.

Response: Thank you for pointing this out. In this study, MAT data were sourced either from the original literature or the climatic database at <http://www.worldclim.org> using coordinate information (please see line 333-335). The lowest MAT data point (-18.7°C) was collected from Alexandra Fiord (78°53'N, 75°47'W) on the eastern coast of Ellesmere Island, Nunavut, Canada (Hudson et al., 2011). Information about these low-temperature points is available in the supplementary materials and data files provided by Craine et al. (2018) (<https://doi.org/10.5061/dryad.v2k2607>). Although these extremely low MAT points represent a small fraction of the overall data, we have thoroughly checked this information to avoid affecting the overall relationships. We have revised the relevant sections of the manuscript accordingly (please check line 30, 192).

The current title is not very informative. I would suggest changing it to “A new framework to estimate soil nitrogen contributions to global terrestrial plants”, or “Novel insights of soil nitrogen contributions to global terrestrial plants”, or something similar.

Response: This is very helpful suggestion. Based on your suggestion, we have revised the title to " Global distribution and drivers of relative contributions among soil nitrogen sources to terrestrial plants" which better encapsulates the core findings of this study.

Line 228 I don't think nitrate leaching has strong isotope fractionation. Please double-check this statement.

Response: Thank you for pointing this out. We have revised this sentence to be “The openness of the soil N cycle would further increase the production and losses of ¹⁵N-depleted N species, such as the N-containing gases (e.g., NO, N₂O, N₂, NH₃)^{22,25,35}.”. Please check line 222-225.

More importantly, the language of the manuscript could be further polished to enhance readability and the impact it deserves.

Response: Thank you for your suggestion. Before resubmission, the English of our manuscript was corrected by our co-authors E. N. Jack Brookshire and Avery W. Driscoll. They are from the Department of Land Resources and Environmental Sciences at Montana State University and the Department of Soil and Crop Sciences at Colorado State University, respectively.

Minor issues:

Line 166 “Figs.” should be “Fig.”

Line 202 “Figs.” should be “Fig.”

“evidnces” in “providing new methods and evidences for evaluating plant N-use mechanisms and N cycle of terrestrial ecosystems.” should be “evidence”.

Response: Thanks. These have been corrected and checked carefully through in the revised manuscript.

References

Craine, J.M. et al. Isotopic evidence for oligotrophication of terrestrial ecosystems. *Nat. Ecol. Evol.* 2, 1735–1744 (2018).

Hudson, J.M.G., Henry, G.H.R., Cornwell W.K. Taller and larger: shifts in Arctic tundra leaf traits after 16 years of experimental warming. *Global Change Biol*, 17, 1013-102 (2011).